# Crop Cultivation Efficiency and GHG Emission: SBM-DEA Model with Undesirable Output Approach

**Tomasz Żyłowski *** [ID] **and Jerzy Kozyra**

Institute of Soil Science and Plant Cultivation—State Research Institute in Puławy, 24-100 Puławy, Poland; kozyr@iung.pulawy.pl
* Correspondence: tzylowski@iung.pulawy.pl

**Abstract:** Crop production relies on the use of natural resources and is a source of greenhouse gas (GHG) emissions. The present study uses survey data from 250 Polish farms to investigate the eco-efficiency of three main crops: winter wheat, winter triticale, and winter oilseed rape. First, the slack-based Data Envelopment Analysis (SBM-DEA) model with undesirable output (GHG emissions) was applied. In the next step, the Generalized Additive Model for Location, Scale and Shape (GAMLSS) was used to explain the efficiency scores. The calculated GHG emissions per hectare of crop were 1.9 tCO$_2$e, 3.2 tCO$_2$e, and 4.3 tCO$_2$e for winter triticale, wheat, and oilseed rape, respectively. Fully efficient farms used significantly less fertilizer (13.6–29.3%) and fuel (16.6–25.3%) while achieving higher yields (14.4–23.2%) and lower GHG emissions per hectare (10.8–17.7%). In practice, this means that efficient farms had a 20–32% lower carbon footprint per kilogram of yield than inefficient farms, depending on the crop. It was also shown that increasing the size of the cultivated area contributed to improving efficiency scores, while no conclusive evidence was found for an influence of economic size or farm type on their performance. Weather conditions had a significant impact on the efficiency score. In general, higher temperatures and precipitation in spring had a positive effect on efficiency, while an opposite relationship was observed in summer.

**Keywords:** carbon footprint; eco-efficiency; GAMLSS model; winter wheat; winter triticale; winter oilseed rape

## 1. Introduction

Modern crop production in developed countries is based on high resource use, but the increase in productivity is associated with an ongoing rise of undesirable environmental impacts, in particular, greenhouse gas emissions. Globally, agriculture accounts for about 12% of GHG emissions, while food production (the entire supply chain from the production of inputs to the disposal of waste) is estimated to account for between 21 and 37% of total emissions [1]. The objectives of the EU's Common Agricultural Policy for 2023–2030 are to ensure food security while reducing the pressure of agriculture on the environment, in particular, to prevent climate change [2]. The EU Green Deal targets include a 50% reduction in nutrient losses, which is expected to lead to at least a 20% reduction in fertilizer use by 2030 without deterioration in soil fertility. It also assumes a 50% reduction in the use of chemical pesticides [3]. In general, all of these policies aim at making crop production more efficient and maintaining (or even increasing) outputs (yields) while reducing input consumption and environmental impacts of cultivation. This is in line with the definition of eco-efficiency, which has three main objectives: reducing the use of resources (energy, materials, water and land), mitigating the impact on nature (GHG emissions, pollution), and increasing the value of the product (this allows to meet the functional need of the customer with fewer materials and resources) [4].

There are a number of methods for measuring agricultural productivity in a broader context, addressing not only economic but environmental impacts adopted from econometrics. The following approaches may be mentioned: indices, parametric models, and linear

programming methods, such as Data Envelopment Analysis (DEA). The advantage of the non-parametric DEA method is that it does not require specific assumptions about distributions of inputs and outputs. Additionally, it can deal with multiple outputs simultaneously, while parametric models can maximize a single output [5]. DEA is the most commonly used linear programming method to measure efficiency in a given set of decision-making units (DMUs). Essentially, this approach identifies efficiency frontiers representing the best possible practices, and the distance between other DMUs and these frontiers is a measure of their relative eco-efficiency [6]. The measured efficiency scores are in interval (0; 1), where a value of unity means that the DMU is fully efficient. There have been many applications of DEA in agricultural systems [7–11]. Various approaches have been used to introduce undesirable results into DEA models, such as ignoring them or treating them as inputs or transforming the data. All of these techniques have their drawbacks, as they change the physical flows, as well as cause difficulties in interpretation and the need to retransform the analysis results. One of the many possible representations of crop production system in the DEA model is taking into account physical inputs and output (yield), as well as undesirable output such as GHG emissions. Pishgar–Komleh et al. (2020) used winter wheat cultivation as an example to compare different approaches of introducing undesirable results into the DEA model [12]. The strength of the slack-based model (SBM–DEA) with undesirable outputs, described by Tone (2003) [13], is that it represents a cropping system without the need to recalculate results or substitute inputs with outputs. Stremikis and Mahyar (2021) analyzed 58 publications using SBM–DEA models with undesirable outputs in agriculture published between 2000 and 2020 and their theoretical and empirical implications for Green Productivity policies [5]. The SBM–DEA model with undesirable outputs was adopted and integrated with Life Cycle Analysis (LCA) results from 10 dairy farms in Umbria (Italy) to estimate their environmental efficiency and emission reduction potential. Dong et al. (2018) [10] used this approach to investigate the efficiency of crop production systems in Zhejiang Province (China). The combination of environmental LCA and DEA was applied to evaluate the eco-efficiency of 169 wheat farms in the north of Iran [14].

The main disadvantage of DEA's model is that it does not provide a direct answer to the question of why DMUs are effective or ineffective. Rather, it allows for the identification of quantitative differences between inputs or outputs for efficient and non-efficient units. Moreover, this approach does not allow a direct assessment of the impact of additional variables (which cannot be included in the DEA model for many reasons). In order to assess the impact of habitat and organizational factors on the efficiency score, a two-stage approach can be applied in which the efficiency scores are used as the dependent variable in the second stage of the analysis. Since the scores are in the range (0; 1), and their distribution has the mass point at unity, the standard regression models (OLS) are generally not appropriate [15]. The use of censored regression models (Tobit) is also questioned [16]. A truncated distribution function and double bootstrap approach described by Simar and Wilson (2007) allows for the inclusion of environmental variables in the analysis and consistent inference within DEA models estimating and explaining efficiency scores while providing standard errors and confidence intervals [17]. This approach was used by Picazo–Tadeo et al. (2011) to explain the causes of inefficiency, taking into account the age, education, and training level of the farmer, as well as the percentage of income from agriculture and the area subject to agri-environmental payments [18]. Quantile regression is also adopted to explain the causes of ineffectiveness. Chidmi et al. (2010) used quantile regression in the second stage analysis to assess the impact of farm size, income, government payments, and non-family labour on dairy farms in Wisconsin (USA) [19].

In this study, the Generalized Additive Models for Location, Scale, and Shape (GAMLSS) was applied to assess the impact of environmental and organizational variables on the efficiency score of cultivation. Due to the distribution of the DEA efficiency results in the interval (0; 1), it is possible to choose the mixed distribution of the explanatory variable in this range. The GAMLSS model was used, inter alia, to evaluate the rice yield in the Banyuwangi province [20] and evaluate the distribution of the rent–price relationship of

agricultural land in Germany [21]. There are no known papers in which the GAMLSS model was used to explain the results of the DEA method, which is the novelty of this study.

The study will analyze the eco-efficiency of the cultivation of main crops in Poland: winter cereals (wheat and triticale) and winter oilseed rape. These are the three crops with the largest sown area in Poland cultivated for different purposes [22]. Wheat is mainly grown for human consumption; triticale is used predominantly for animal feed (Poland is the largest producer of triticale in the world); oilseed rape is an industrial crop, and its oil is used as a bio-component for diesel fuel and for human consumption. A slack-based DEA model is used, with greenhouse gas emissions as the undesirable result. Then, to test the hypothesis that GHG emissions can be reduced, the differences between efficient and inefficient farms are compared. We will also investigate whether farm size, type, cropping area, weather patterns, and other factors have a significant impact on efficiency scores. These results could provide valuable assistance to farmers in identifying hot spots and improving production techniques that may lead to excessive input use and GHG emissions.

## 2. Materials and Methods

### 2.1. Data and Data Curation

The data used in the study are obtained from surveys conducted in 2015/2016 and 2016/2017 seasons on 250 farms across Poland. The surveyed farms belonged to three types of production, according to Farm Accountancy Data Network (FADN) classification: arable crops, dairy cows, and pigs. The collected data include detailed information on tillage (type of treatment, machines used, machine operating time, diesel used), mineral and organic fertilizer application, pesticide use, seeds, and yield. The contents of nitrogen, phosphorus, and potassium in organic fertilizers used in the farms were disaggregated by the type of fertilizer (slurry, manure, and liquid manure) and animal species (cattle, pigs) [23]. The content of active nutrients per year of application was then calculated using the corresponding fertilizer equivalents according to Polish legislation [24].

To detect outlier observations in multivariate data, the density-based spatial clustering of application with noise (DBSCAN) algorithm proposed by Ester [12,25], implemented in the dbscan (v 1.1-11) package, was used [26]. This enabled the identification of outlier observations in low-density areas in multivariate data and did not require prior knowledge of the number of clusters [26]. Application of this method resulted in rejections of 5% (12 farms), 4.2% (13 farms), and 4.4% (8 farms) for winter triticale, winter wheat, and winter oilseed rape, respectively. The overview of the input–output inventory after data curation is presented in Table 1.

**Table 1.** Descriptive statistics of input and output data for chosen crops (all values, except number of farms, refer to 1 ha of cultivation).

| Input/Output | Unit | Winter Wheat | Winter Triticale | Winter Oilseed Rape |
|---|---|---|---|---|
| Number of farms | | 293 | 230 | 174 |
| Inputs | | | | |
| 1. Seed | kg | 200.2 ($\pm$32.1) | 199.9 ($\pm$30.9) | 3.3 ($\pm$0.8) |
| 2. Diesel fuel | l | 109.1 ($\pm$37.6) | 96.9 ($\pm$36.8) | 111.0 ($\pm$35.1) |
| 3. Machinery | h | 9.5 ($\pm$3.3) | 9.5 ($\pm$3.4) | 8.8 ($\pm$2.9) |
| 4. Total NPK | kg | 238.9 ($\pm$107.4) | 262.1 ($\pm$131.1) | 308.3 ($\pm$121.7) |
| Mineral N | kg | 124.7 ($\pm$61.9) | 96.6 ($\pm$46.7) | 172.4 ($\pm$55.1) |
| Mineral $P_2O_5$ | kg | 27.1 ($\pm$31.3) | 30.0 ($\pm$24.8) | 43.6 ($\pm$44.1) |
| Mineral $K_2O$ | kg | 44.7 ($\pm$45.4) | 46.6 ($\pm$ 22.0) | 63.6 ($\pm$43.6) |
| Natural N | kg | 11.0 ($\pm$22.8) | 22.0 ($\pm$27.3) | 7.9 ($\pm$20.8) |
| Natural $P_2O_5$ | kg | 9.3 ($\pm$20.4) | 17.1 ($\pm$22.8) | 7.1 ($\pm$19.4) |
| Natural $K_2O$ | kg | 20.4 ($\pm$48.2) | 49.9 ($\pm$67.0) | 13.8 ($\pm$38.9) |
| 5. Pesticides | kg a.i. | 3.4 ($\pm$1.8) | 2.6 ($\pm$1.8) | 4.3 ($\pm$1.6) |
| Output (grain) | t | 5.8 ($\pm$1.4) | 4.6 ($\pm$1.2) | 3.0 ($\pm$0.8) |

A detailed description of the auxiliary variables is contained in Table 2. The analyzed farms were classified according to the Polish FADN system with production types (TF8): plant cultivation (crop), dairy cattle (dairy), and pigs (pig), and to three economic size classes (ES6): small (8–25 thousand EUR), medium (25–100 thousand EUR), large (100–500 thousand EUR). The farms are located in four FADN regions (across Poland): Region A (785) includes Pomorze and Mazury; B (790) Wielkopolska and Śląsk; C (795) Mazowsze and Podlasie; D (800) Małopolska and Pogórze. Soil quality was aggregated into three levels: good, medium, and poor. Climate data were gathered from the EOBS network [27] and processed with the climate data operator (CDO, v. 2.0.4 ) software [28].

**Table 2.** Explanatory variables used in the analyses.

| Variable Name | Description | Range |
| --- | --- | --- |
| area | Size of arable field [ha] | [0.2; 50] |
| temp_autumn | Mean of autumn air temperature (IX-XI) [°C] | [9.22; 12.97] |
| prec_autumn | Sum of autumn precipitation [mm] | [62.90; 215.6] |
| temp_winter | Mean of winter air temperature (XII-II) [°C] | [−1.37; 2.17] |
| prec_winter | Sum of winter precipitation [mm] | [80.30; 236.2] |
| temp_spring | Mean of spring air temperature (III-V) [°C] | [7.49; 10.43] |
| prec_spring | Sum of spring precipitation [mm] | [72.50; 261.8] |
| temp_summer | Mean of summer air temperature (VI-VII) [°C] | [16.01; 19.44] |
| prec_summer | Sum of summer precipitation [mm] | [66.00; 292.1] |
| soil | Soil class | {good *, medium, poor} |
| type | The main type of production (FADN) | {crop *, dairy, pig} |
| econ_class | Economic size of farm (FADN) | {big *, medium, small} |
| organic_fert | Use of natural fertilizers (manure/slurry) in a given season | {no *, yes} |
| year | Season of cultivation (2015/2016 or 2016/2017) | {2016 *, 2017} |
| fadn | Belongs to Polish FADN region | {A *, B, C, D} |
| residue_collected | Is residue collected? | {no *, yes} |

The ranges of quantitative variables are detailed in square brackets, qualitative—in curly brackets. Reference levels for categorical variables (for modelling) are marked *.

### 2.2. Carbon Footprint Evaluation

A carbon footprint is defined as the balance (emissions or removals) of greenhouse gases caused by a product and expressed in carbon dioxide equivalent $CO_2e$ [29]. The scope of the analysis includes upstream emissions (input production), as well as direct on-farm emissions from input use and GHG emissions from the soil. Two functional units: 1 hectare of land cultivated (CF_ha) and 1 kg of yield (CF_kg), were adopted for further analysis. Farmland $CO_2$ and $N_2O$ emissions were calculated using the refined IPCC methodology [30]. The ammonia volatilization factor from mineral fertilizers disaggregated into nitrogen forms (nitrate, ammonium, ammonium-nitrate, and amide) was adopted. In the absence of data from each field, the dry matter content in the crop yield, the ratio of the main yield (grain) to the secondary yield (straw), and nitrogen content were taken from the National Inventory Report [31] as the ones that best correspond to Polish conditions. For the analyses, it was assumed that for $N_2O$, the global warming potential over a 100-year time horizon is equal to 298 [32].

To calculate the emissions related to the production of inputs, emission coefficients from the Biograce project [33] and ecoinvent database version 3.4 [34] were used. The emissions from the use of machines were adopted separately for the tractors, agricultural machines, and combine harvesters [35,36]. It was assumed that the carbon footprint of triticale seeds is equal to that of wheat. In accordance with the convention adopted in most LCI databases and the Biograce project, the carbon footprint of organic fertilizers has been related to livestock production and has been assumed to be zero [33,34,37]. Similarly, due to the study period and the lack of data on soil organic matter changes, the balance of soil organic carbon was assumed to be zero. A detailed summary of input emission factors is shown in Table 3.

**Table 3.** GHG emission factors from the production of the inputs used.

| Emissions Related to Agricultural Inputs | Unit | kg CO$_2$e/Unit | Source | Comments/Details |
|---|---|---|---|---|
| Inputs | | | | |
| Urea-based fertilizer | kg N | 3.17 | [33] | |
| Ammonium based | kg N | 1.62 | [33] | Ammonium sulphate |
| Nitrate based | kg N | 6.34 | [33] | CAN |
| Ammonium nitrate | kg N | 6.21 | [33] | |
| P$_2$O | kg P$_2$O$_5$ | 1.01 | [33] | |
| K$_2$O$_5$ | kg K$_2$O | 0.58 | [33] | |
| CaO | kg CaO | 0.13 | [33] | |
| Pesticides | kg | 10.97 | [33] | |
| Diesel fuel | l | 3.15 | [33] | |
| Machinery | | | | |
| Tractor | kg | 7.78 | [34] | Tractor, 4-wheel, agricultural {RoW}\|production\|APOS, U |
| Equipment | kg | 5.56 | [34] | Agricultural machinery, unspecified {RoW}\|production\|APOS\|U |
| Harvester | kg | 6.66 | [34] | Harvester{RoW}\|production\|APOS\|U |
| Seeds | | | | |
| winter wheat/triticale | kg | 0.87 | [34] | Wheat seed, for sowing {GLO}\|market for\|APOS, U |
| winter oilseed rape | kg | 1.23 | [34] | Rape seed, for sowing {RoW}\|market for rape seed, for sowing\|APOS, U |
| Manure/Slurry | t/m$^3$ | 0.00 | [33] | |

*2.3. Efficiency Measurement Using SBM-DEA Model with Undesirable Output*

Data envelopment analysis is a widely known and used nonparametrical technique for assessing the relative efficiency of DMUs [38]. Over the years since Charnes et al. (1978) introduced the CCR model [39], many more advanced models have been developed for different types of applications [40]. In general, the most important parameters of DEA models are orientation and assumption of returns to scale. Input-oriented basic models aim to minimize inputs while producing at least the given output levels, while output-oriented models aim to maximize output levels while producing at most the given input consumption. The CCR model measures the technical efficiency (TE) of DMUs under the constant return to scale (CRS) assumption while efficiency under variable return to scale (VRS) condition is known as pure technical efficiency (PTE) or local efficiency. The effect of DMU size on efficiency could be calculated by scale efficiency (SE) as the ratio of TE to PTE. The assumption of constant returns to scale means that all outputs and all inputs increase (or decrease) at the same rate, while the assumption of variable returns to scale (VRS) enables some of DMUs to have a constant, decreasing, or increasing return to scale. Increasing return to scale (IRS) is when a proportional increase in all the inputs results in a more than proportional increase in output, while decreasing return to scale (DRS) means proportional increase in all the inputs results in a less than proportional increase in output [41].

In this study, an SBM-DEA input-oriented model with undesirable output (GHG emissions) was applied to assess the relative efficiency of crop cultivation. A detailed description of the model can be found in the works of Karou Tone [13,42]. The main advantage of this approach is that it does not require changing the physical relationship between inputs and outputs (moving undesirable outputs to inputs or undesirable inputs to outputs) or transforming variables (which may lead to unexpected results) [43]. The SBM-DEA model, unlike the radial DEA models, works directly with slacks (excess input and shortfalls in the output) and abandons the hypothesis of proportional variable changes. Technical, pure technical, and scale efficiency for three crops were estimated. The following variables were used as inputs: total amount of nutrients (sum of nitrogen, phosphorus,

and potassium from mineral and natural fertilizers) [kg], diesel fuel [l], pesticides [kg a.i], seeds [kg], machinery [minutes]. The GHG emissions [kg $CO_2$e] (undesirable output) and the yield (grain) [kg] were deployed as outputs. The deaR (v. 1.4) package [44] was applied for the calculations. PTE scores were adopted as the dependent variable in further models estimation due to the fact that increasing inputs in crop cultivation does not necessarily lead to proportional changes in results [45].

*2.4. Explaining Efficiency*

　　Since the Data Envelopment Analysis method does not allow for direct determination of the causes of efficiency/inefficiency, the generalized additive model for location, scale and shape (GAMLSS), introduced by Stasinopoulos et al. (2007), was used to determine them [46]. The study used a one-inflated beta distribution (BEINF1), which allows fitting a beta distribution with $\mu$ (mean) and $\sigma$ (standard deviation) parameters on (0,1) with extra point probabilities ($\nu$ parameter) at unity. It was implemented as a three-parameter distribution BEINF1($\mu$, $\sigma$, $\nu$), using logit ($\mu$), log ($\sigma$), logit ($\nu$) link functions, respectively [47]. This approach allows for the evaluation of not only the expected mean value, but also the scale parameter (variability) and the shape parameter (related to the probability of the unity value). The final models were obtained using an independent variable selection procedure based on the Generalized Akaike Information Criterion (GAIC). In the selected version of the algorithm, the procedure starts with the empty model (containing only constants) and performs a forward stepwise selection for subsequent model parameters. The gamlss (v. 5.4-10) package [46] was used to estimate the GAMLSS models.

　　To assess the impact of the habitat, quantitative variables describing the weather for the seasons of the year were used (temperature, precipitation). Dummy variable *year* is used as a proxy of differences between growing seasons that are impossible to capture in another way. The effect of soil quality is determined by the soil variable. Structural/organizational variables are represented by cultivation area, type of production, and economic size of the farm (econ_class). These variables were used, inter alia, in the work of Kaditi [48]. The geographical location broken down into FADN regions is determined by the fadn variable. It can be treated as a proxy for differences resulting from tradition and the agricultural structure, as well as input intensity. The effect of the use of organic fertilizers on PTE was investigated using the variable organic_fert. A detailed description of the used variables was provided above in Table 2.

　　The Kruskal–Wallis non-parametric tests were used for comparison between groups of an independent variable on a continuous variable; posthoc comparison between groups applied the Wilcoxon rank sum test. Pearson's Chi-squared test was carried out to determine whether there is a statistically significant difference between categorical variables. In the following sections, expression "significant differences" are referred to test results for which *p*-value < 0.05.

**3. Results**

*3.1. Carbon Footprint*

　　The results of the analyses indicate that there are significant differences in the level of the carbon footprint of the studied crops (Figure 1). The largest carbon footprint per hectare was evaluated for winter oilseed rape 4271 ($\pm$594) kg $CO_2$e ha$^{-1}$, followed by wheat 3197 ($\pm$685) kg $CO_2$e ha$^{-1}$ and triticale 1915 ($\pm$520) kg $CO_2$e ha$^{-1}$. The results obtained indicate that the carbon footprint of 1 kg of yield is 1.57 ($\pm$0.58), 0.58 ($\pm$0.18), 0.45 ($\pm$0.20) kg $CO_2$e kg$^{-1}$ for oilseed rape, wheat, and triticale, respectively. The largest contributors to the carbon footprint are GHG emissions (mainly $N_2O$) from the soil at a share of 65%, 57%, and 40% for winter oilseed rape, winter wheat and winter triticale, respectively (Figure 2). The magnitude of these emissions is controlled by the amount of nitrogen entering the soil in the form of fertilizer application, crop residues, or mineralization of soil organic matter. This is confirmed by the strong correlation between the amount of total nitrogen fertilization and soil GHG emissions of 81.6% for wheat, 85.5% for triticale, and 84% for oilseed rape.

Significant differences were also confirmed between the levels of total nitrogen fertilization of the investigated crops. Due to the methodology applied, the use of organic fertilizers is "promoted" (emissions from their production are attributed to the livestock production system; only field $N_2O$ emissions are accounted), which may explain the differences in the carbon footprints of the crops studied: organic fertilizers were used by 14% of winter oilseed rape farms, 23% of wheat farms, and 46% of triticale farms and accounted for 3.6, 8.3, and 16.5% of the nitrogen input from fertilizers, respectively. Other sources of greenhouse gas emissions are the production of mineral fertilizers, which accounts for 22.8% to 28.7%, and the production and combustion of diesel and the use of agricultural machinery, which together account for 10.4% (oilseed rape), 13.7% (wheat), and 20.8% (triticale). Plant protection products contribute no more than 1.5% for all crops, while seed production has no significant impact on GHG emissions for oilseed rape; its contribution is relatively higher for cereals. It should be noted that, on average, about 3.3 kg of rape seed is used compared to about 200 kg of cereal seed.

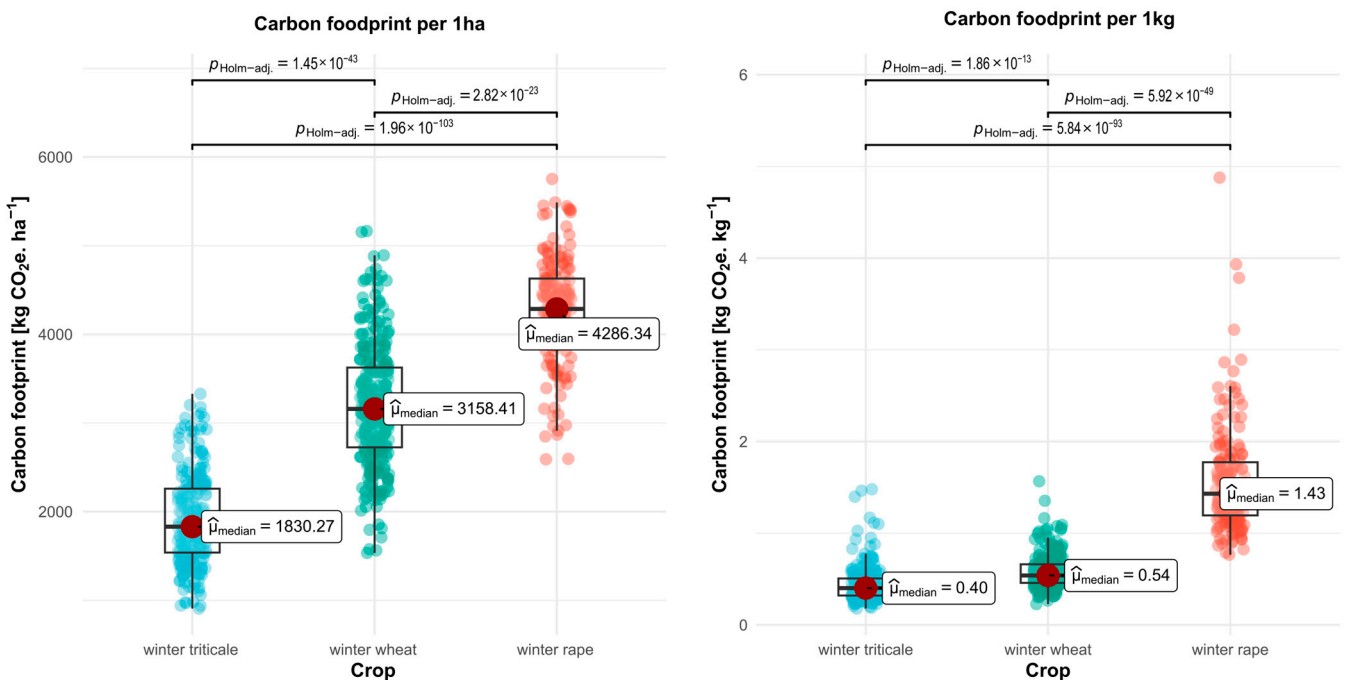

**Figure 1.** Differences in the carbon footprint of 1 ha of cultivation and 1 kg of yield for the analyzed crops.

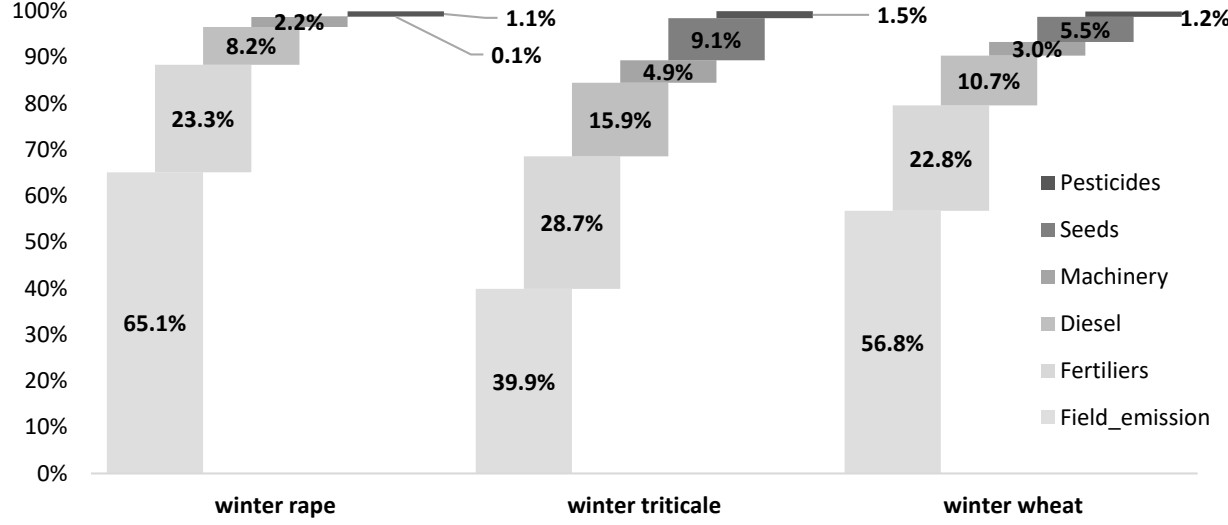

**Figure 2.** Share of different emission sources in carbon footprint of the studied crops.

Detailed analyses of the carbon footprint for each crop by economic class and type of farm are presented in Tables 4 and 5. There were no differences in CF of 1 kg of yield between farms grouped according to economic size for all analyzed crops. On the other hand, there are differences in the carbon footprint per hectare of cereals cultivation between big and other farms (variable econ_class). Similar relationships between input levels are confirmed; big farms growing winter triticale and winter wheat use more NPK fertilizer and achieve higher yields. At the same time, big farms use more diesel fuel with less machinery time, which may indicate that they have more modern equipment.

**Table 4.** Key inputs and carbon footprint characteristics of farms by economic class. The same letters indicate no significant statistical difference between variables. For clarity, only mean values are shown.

| Crop/ Variable | Econ_Class | Area [ha] | Yield [t ha$^{-1}$] | NPK_Total [kg ha$^{-1}$] | Fuel [kg ha$^{-1}$] | Machinery [h ha$^{-1}$] | CF_ha [kg CO$_2$e ha$^{-1}$] | CF_kg [kg CO$_2$e kg$^{-1}$] |
|---|---|---|---|---|---|---|---|---|
| winter rape | big | 25.9 a | 3.03 a | 317 a | 116 a | 8.23 a | 4315 a | 1.51 a |
| | medium | 14.3 b | 2.98 a | 297 a | 106 a | 9.09 ab | 4203 a | 1.57 a |
| | small | 6.29 c | 2.56 a | 323 a | 110 a | 10.4 b | 4401 a | 1.95 a |
| winter triticale | big | 11.9 a | 5.11 a | 301 a | 114 a | 9.25 a | 2121 a | 0.428 a |
| | medium | 6.2 b | 4.51 b | 257 b | 90.4 b | 9.44 a | 1849 b | 0.458 a |
| | small | 3.12 c | 4.09 b | 202 c | 83.1 c | 10.1 a | 1717 b | 0.447 a |
| winter wheat | big | 30.4 a | 6.07 a | 278 a | 125 a | 8.54 a | 3424 a | 0.586 a |
| | medium | 12 b | 5.85 ab | 226 b | 103 b | 9.85 b | 3131 b | 0.562 a |
| | small | 4.86 c | 5.08 b | 190 b | 94.9 b | 11 c | 2850 c | 0.623 a |

**Table 5.** Key inputs and carbon footprint characteristics of farms by type. The same letters indicate no significant statistical difference between variables.

| Crop/ Variable | Type | Area [ha] | Yield [t ha$^{-1}$] | NPK_Total [kg ha$^{-1}$] | Fuel [kg ha$^{-1}$] | Machinery [h ha$^{-1}$] | CF_ha [kg CO$_2$e ha$^{-1}$] | CF_kg [kg CO$_2$e kg$^{-1}$] |
|---|---|---|---|---|---|---|---|---|
| winter rape | crop | 23.7 a | 2.91 a | 287 a | 106 a | 8.54 a | 4231 a | 1.6 a |
| | dairy | 7.9 b | 3.04 a | 331 ab | 126 a | 8.8 a | 4420 a | 1.56 a |
| | pig | 8.53 b | 3.11 a | 363 b | 122 a | 9.5 a | 4350 a | 1.51 a |
| winter triticale | crop | 9.44 a | 4.35 a | 200 a | 82.2 a | 7.51 a | 1918 a b | 0.472 a |
| | dairy | 4.34 b | 4.47 a | 264 b | 93.1 a | 9.73 b | 1809 a | 0.444 a |
| | pig | 9.21 a | 4.93 b | 295 b | 108 b | 10.4 b | 2005 b | 0.434 a |
| winter wheat | crop | 28 a | 6.02 a | 223 a | 106 a | 8.97 a | 3297 a | 0.584 a |
| | dairy | 4.27 b | 5.3 b | 205 a | 103 a | 10.2 b | 2909 b | 0.574 a |
| | pig | 7.21 c | 5.9 a | 291 b | 120 a | 10.2 a b | 3220 a | 0.565 a |

Differences in the carbon footprint per 1 kg of yield due to farm type (variable type) for individual crops were not confirmed. The results indicate that dairy farms have a lower (winter wheat) or similar carbon footprint per 1 hectare of cultivation compared to crop farms (winter triticale); in the case of oilseed rape, no differences were confirmed. For all cultivated crops, pig farms applied the highest amount of NPK fertilizer and achieved yields comparable to those of crop farms.

### 3.2. SBM-DEA Efficient vs. Non-Efficient Farms

The SBM-DEA model classifies farms into relatively efficient and inefficient ones. Table 6 shows the detailed results of the efficiency analysis. Note that technical efficiency score equal to unity (TE = 1) means that such farm is also scale efficient (SE = 1), which follows directly the definition of scale efficiency. No differences were found between the cultivated crops in proportion of DMUs having pure technical efficiency score equal to unity (PTE = 1), with the percentage share of fully efficient farms ranging from 11.6 to 15.2. The results for the type of scale efficiency indicate that the majority of farms are not operating at the optimum level of production, but at the same time, the relatively high average scale efficiency scores show that farms are operating close to their optimum size.

**Table 6.** Descriptive statistics of efficiency and scale of operations by studied crops. Mean values and standard deviation (in parentheses) are shown for numeric variables; value and share in relation to total number of farms (in parentheses) for integer variables.

| Item | Winter Rape | Winter Triticale | Winter Wheat |
|---|---|---|---|
| Total number of farms | 174 | 230 | 293 |
| TE = 1 | 12 (6.9%) | 20 (8.7%) | 20 (6.8%) |
| PTE = 1 | 25 (14.4%) | 35 (15.2%) | 34 (11.6%) |
| TE score | 0.49 (±0.20) | 0.47(±0.20) | 0.50 (±0.18) |
| PTE score | 0.56 (±0.22) | 0.53 (±0.23) | 0.56 (±0.21) |
| SE score | 0.88 (±0.13) | 0.90 (±0.13) | 0.90 (±0.12) |
| Type of scale efficiency | | | |
| Increasing Return to Scale (IRS) | 80 (46.0%) | 66 (28.7%) | 184 (62.8%) |
| Constant Return to Scale (CRS) | 12 (6.9%) | 20 (8.7%) | 20 (6.8%) |
| Decreasing Return to Scale (DRS) | 82 (47.1%) | 144 (62.6%) | 89 (30.4%) |

A detailed analysis of the effects of return to scale on the cultivation of the crops is presented in Table 7. The average optimum field size for winter oilseed rape is 17.6 ha and results in the highest yield with the lowest N fertilizer input (and thus the lowest emissions per 1 ha of the area and 1 kg of yield). A similar relationship was found for triticale and winter wheat, where the optimal field size was 8.2 and 16.4 ha, respectively. Farms with increasing returns to scale are characterized by a smaller cultivation area and a lower yield. Farms in decreasing returns to scale grow oilseed rape and wheat on larger area (for triticale this relationship is not statistically significant) and with a higher level of nitrogen fertilization (resulting in higher GHG emissions). The results of the analyses indicate that IRS farms should increase the scale of production and should reduce the cultivation area.

**Table 7.** Characteristics of the main parameters affecting the efficiency and carbon footprint of the investigated farms in relation to the type of return to scale. Same letters indicate similar levels of variable.

| Crop/ Variable | VRS | Area [ha] | Yield [t ha$^{-1}$] | N_min [kg ha$^{-1}$] | N_org [kg ha$^{-1}$] | CF_ha [kg CO$_2$e ha$^{-1}$] | CF_kg [kg CO$_2$e kg$^{-1}$] |
|---|---|---|---|---|---|---|---|
| winter rape | Constant | 17.6 a | 3.8 a | 107.8 a | 0.6 a b | 3535.0 a | 1.13 a |
| | Decreasing | 31.4 b | 3.1 b | 178.0 b | 4.4 a | 4346.2 b | 1.50 b |
| | Increasing | 6.6 c | 2.8 c | 176.3 b | 12.4 b | 4304.5 b | 1.71 c |
| winter triticale | Constant | 8.2 a | 5.5 a | 80.5 a | 13.4 a b | 1596.4 a | 0.30 a |
| | Decreasing | 9.4 a | 4.7 b | 101.6 b | 26.7 a | 2009.4 b | 0.45 b |
| | Increasing | 3.2 b | 4.2 c | 90.5 a b | 14.2 b | 1805.4 c | 0.49 b |
| winter wheat | Constant | 16.4 a | 7.1 a | 89.0 a | 5.5 a b | 2705.6 a | 0.40 a |
| | Decreasing | 41.7 b | 6.5 a | 151.6 b | 2.4 a | 3476.0 b | 0.55 b |
| | Increasing | 5.5 c | 5.4 b | 115.6 a | 15.7 b | 3114.7 c | 0.61 c |

Table 8 compares efficient and inefficient (PTE) farms in terms of the inputs and outputs. The results indicate that efficient farms cultivating triticale and wheat use lower levels of inputs, emit less greenhouse gases, and achieve higher yield. Similarly, the results indicate that there are significant differences in the carbon footprint of the crop cultivation related to 1kg yield for all crops. Mean values of CF for efficient and non-efficient farms for oilseed rape are 1.29 vs. 1.62 kgCO$_2$e kg$^{-1}$ ($X^2_{K-W}$ = 23.945, *p*-value = 9.915 × 10$^{-7}$), wheat 0.44 vs. 0.59 kgCO$_2$e kg$^{-1}$ ($X^2_{K-W}$ = 28.784, *p*-value = 8.092 × 10$^{-8}$), and triticale 0.32 vs. 0.47 kgCO$_2$e kg$^{-1}$ ($X^2_{K-W}$ = 28.573, *p*-value = 9.025 × 10$^{-8}$). In practice, this means that efficient farms emit 20%, 32%, 27% less GHG per kg of yield than inefficient farms, for oilseed rape, triticale, and wheat, respectively.

**Table 8.** Input and output levels for efficient and inefficient (PTE) farms derived from the SBM-DEA model (all values refer to 1 ha of cultivation). The column Diff was calculated as the ratio of the difference between the value of the non-efficient and efficient farms and the value of the efficient farms, expressed as a percentage. A positive value indicates that the efficient farms use less inputs and emit less GHG, and negative values for the yield variable indicate that they achieve higher yields. Asterisks in the Diff column indicate statistically significant differences (Kruskall–Wallis test, *p*-value < 0.05).

| Item | Oilseed Rape | | | Triticale | | | Wheat | | |
|---|---|---|---|---|---|---|---|---|---|
| | **PTE = 1** | **PTE < 1** | **Diff** | **PTE = 1** | **PTE < 1** | **Diff** | **PTE = 1** | **PTE < 1** | **Diff** |
| NPK total [kg] | 249.0 | 318.3 | 21.8% * | 274.4 | 194.0 | 29.3% * | 209.6 | 242.6 | 13.6% * |
| Diesel [L] | 89.0 | 114.7 | 22.4% * | 99.4 | 82.9 | 16.6% * | 84.1 | 112.6 | 25.3% * |
| Machinery [hours] | 7.5 | 9.0 | 17.1% * | 9.8 | 7.7 | 21.4% * | 7.2 | 9.8 | 26.6% * |
| Seeds [kg] | 3.3 | 3.4 | 3.0% | 199.9 | 199.6 | 0.2% | 189 | 201.8 | 6.3% * |
| Pesticides [kg a.i.] | 3.0 | 4.5 | 33.2% * | 2.9 | 1.4 | 51.7% * | 3 | 3.4 | 12.5% |
| CF_ha [kg $CO_2$e] | 3809.3 | 4348.6 | 12.4% * | 1619.6 | 1968.0 | 17.7% * | 2887.7 | 3237.1 | 10.8% * |
| Yield [kg] | 3540.8 | 2874 | −23.2% * | 4532.7 | 5184.6 | −14.4% * | 6891.2 | 5704.2 | −20.8% * |

### 3.3. Factors That Influence Eco-Efficiency

The models assuming the beta one-inflated distribution (BEINF1) was used to assess the impact of environmental and structural variables on the PTE score of crops cultivation. The qualitative impact of the explanatory variables on PTE is shown in Table 9, while detailed estimations for final model parameters are presented in Tables 10–12. Additionally partial effects on the mean value of PTE (mu parameter of BEINF1 distribution) of selected explanatory variables are visualized in Figures 3–5. The quality of the models was examined by analyzing randomized quantile residuals. The randomized quantile residuals of the winter triticale model identified a mean near zero (−0.038), their variance approximated one (0.995) and their coefficient of kurtosis was close to three (2.751). This was also true for the residuals of the winter wheat model (mean = −0.010, variance = 1.034, kurtosis = 3.202), and winter oilseed rape model (mean = 0.018, variance = 1.002, kurtosis = 2.972). Thus, the models fit the data reasonably well, with the residuals of the final models being approximately normally distributed.

**Table 9.** The impact of investigated variables on mean PTE score (estimation of μ model parameter).

| Item | Winter Rape | Winter Triticale | Winter Wheat |
|---|---|---|---|
| area | ↑ * | ↑ | ↑ * |
| temp_autumn | | ↓ * | |
| temp_winter | | | ↓ * |
| prec_winter | ↑ * | | |
| temp_spring | | ↑ * | |
| prec_spring | ↑ * | ↑ | |
| temp_summer | | ↓ | |
| prec_summer | ↓ * | | |
| soil = medium | | ↓ | |
| soil = poor | | ↓ * | |
| organic_fert = yes | | ↓ * | ↓ * |
| year = 2017 | ↑ * | | ↑ * |
| residue_collected = yes | ↑ | | |
| intercrop = yes | | ↑ * | |

The arrows indicate the positive (↑) or negative (↓) impact of the variable on PTE. Asterisks (*) denote statistical significance of the parameter (*p*-value < 0.05).

**Table 10.** Estimation results for final GAMLSS model (winter triticale).

| Winter Triticale Final Model: PTE~f($\mu$, $\sigma$, $\nu$) | | | | | |
|---|---|---|---|---|---|
| logit ($\mu$) | | | | | |
| | Estimate | Std. Error | t-value | Pr(>|t|) | |
| Intercept | 0.955174 | 1.097768 | 0.87 | 0.3853 | |
| area | 0.006279 | 0.006082 | 1.032 | 0.3032 | |
| temp_autumn | −0.19332 | 0.082152 | −2.353 | 0.0196 | * |
| temp_spring | 0.45148 | 0.207186 | 2.179 | 0.0305 | * |
| temp_summer | −0.16958 | 0.135658 | −1.25 | 0.2127 | |
| prec_spring | 0.001588 | 0.001425 | 1.115 | 0.2663 | |
| soil_class = medium | −0.1749 | 0.112606 | −1.553 | 0.1219 | |
| soil_class = poor | −0.29521 | 0.137317 | −2.15 | 0.0327 | * |
| intercrop = yes | 0.208173 | 0.104801 | 1.986 | 0.0483 | * |
| organic_fert = yes | −0.150683 | 0.064945 | −2.32 | 0.0213 | * |
| logit ($\sigma$) | | | | | |
| Intercept | −2.736911 | 0.534855 | −5.117 | $7.12 \times 10^{-7}$ | *** |
| area | $1.51 \times 10^{-2}$ | $1.51 \times 10^{-2}$ | 1.356 | 0.1766 | |
| prec_summer | $3.75 \times 10^{-3}$ | $1.53 \times 10^{-3}$ | 2.451 | 0.0151 | * |
| prec_spring | $4.40 \times 10^{-3}$ | $2.35 \times 10^{-3}$ | 1.875 | 0.0622 | . |
| econ_class = medium | $2.72 \times 10^{-1}$ | $1.52 \times 10^{-1}$ | 1.791 | 0.0748 | . |
| econ_classs = small | $-1.30 \times 10^{-1}$ | $2.06 \times 10^{-1}$ | −0.632 | 0.5284 | |
| type = dairy | $3.43 \times 10^{-1}$ | $1.94 \times 10^{-1}$ | 1.767 | 0.0787 | . |
| type = pig | $4.07 \times 10^{-2}$ | $1.88 \times 10^{-1}$ | 0.217 | 0.8284 | |
| organic_fert = yes | $2.38 \times 10^{-1}$ | $1.40 \times 10^{-1}$ | 1.702 | 0.0903 | . |
| year = 2017 | $-4.30 \times 10^{-1}$ | $2.78 \times 10^{-1}$ | 15463.1 | $<2 \times 10^{-16}$ | *** |
| log ($\nu$) | | | | | |
| Intercept | −1.32892 | 0.53396 | −2.489 | 0.01361 | * |
| area | 0.06408 | 0.01944 | 3.297 | 0.00115 | ** |
| soil_class = medium | −0.87209 | 0.58519 | −1.49 | 0.13769 | |
| soil_class = poor | 0.08487 | 0.66581 | 0.127 | 0.89869 | |
| organic_fert = yes | −1.01697 | 0.43303 | −2.349 | 0.0198 | * |

Cox Snell pseudo $R^2$ = 0.35, Symbols ***, **, *, ˙ denote coefficients which are significant at 0.001, 0.01, 0.05, 0.1 levels, respectively.

**Table 11.** Estimation results for final GAMLSS model (winter wheat).

| Winter Wheat Final Model: PTE~f($\mu$, $\sigma$, $\nu$) | | | | | |
|---|---|---|---|---|---|
| logit ($\mu$) | | | | | |
| | Estimate | Std. Error | t-value | Pr(>|t|) | |
| (Intercept) | −0.085217 | 0.057847 | −1.473 | 0.14186 | |
| area | 0.00856 | 0.002137 | 4.006 | $7.96 \times 10^{-5}$ | *** |
| temp_winter | −0.102088 | 0.038769 | −2.633 | 0.00894 | ** |
| organic_fert = yes | −0.274073 | 0.065262 | −4.2 | $3.62 \times 10^{-5}$ | *** |
| year = 2017 | 0.16694 | 0.062716 | 2.662 | 0.00823 | ** |
| logit ($\sigma$) | | | | | |
| Intercept | 3.63557 | 1.365483 | 2.662 | 0.00822 | ** |
| area | 0.005196 | 0.003506 | 1.482 | 0.13943 | |
| temp_summer | −0.26405 | 0.075841 | −3.482 | 0.00058 | *** |
| econ_class = medium | −0.03608 | 0.129021 | −0.28 | 0.77994 | |
| econ_classs = small | 0.44972 | 0.210833 | 2.133 | 0.03381 | * |
| organic_fert = yes | −0.22761 | 0.13114 | −1.736 | 0.08376 | . |
| log ($\nu$) | | | | | |
| (Intercept) | −4.3938 | 1.266181 | −3.47 | 0.000604 | *** |
| area | 0.048616 | 0.011791 | 4.123 | $4.96 \times 10^{-5}$ | *** |
| prec_spring | 0.013238 | 0.006016 | 2.201 | 0.028595 | * |
| prec_autumn | −0.01913 | 0.008191 | −2.335 | 0.020267 | * |
| type = dairy | 1.593927 | 0.606189 | 2.629 | 0.009036 | ** |
| type = pig | 1.328754 | 0.658671 | 2.017 | 0.044636 | * |
| econ_class = medium | 1.253814 | 0.62203 | 2.016 | 0.044809 | * |
| econ_class = small | 1.40318 | 0.94883 | 1.479 | 0.140328 | |

Cox Snell pseudo $R^2$ = 0,32, Symbols ***, **, *, ˙ denote coefficients which are significant at 0.001, 0.01, 0.05, 0.1 levels, respectively.

**Table 12.** Estimation results for final GAMLSS model (winter oliseed rape).

| | | | Winter Oilseed Rape Final Model: PTE~f(μ, σ, ν) | | |
|---|---|---|---|---|---|
| | | | logit (μ) | | |
| | Estimate | Std. Error | t-value | Pr(>\|t\|) | |
| Intercept | −0.61884 | 0.270816 | −2.285 | 0.023662 | * |
| area | 0.005752 | 0.002054 | 2.8 | 0.005758 | ** |
| prec_winter | 0.00387 | 0.001345 | 2.877 | 0.004586 | ** |
| prec_spring | 0.002801 | 0.000734 | 3.818 | 0.000194 | *** |
| prec_summer | −0.00326 | 0.000946 | −3.448 | 0.000726 | *** |
| residue_colected = yes | 0.161078 | 0.135882 | 1.185 | 0.237665 | |
| year = 2017 | 0.200319 | 0.076398 | 2.622 | 0.009612 | ** |
| | | | logit (σ) | | |
| Intercept | −3.60193 | 2.625933 | −1.372 | 0.1721 | |
| temp_summer | 0.287265 | 0.139291 | 2.062 | 0.0408 | * |
| prec_spring | −0.00559 | 0.002846 | −1.963 | 0.0515 | . |
| prec_summer | −0.00488 | 0.002401 | −2.032 | 0.0439 | * |
| fadn = B | −0.44165 | 0.266573 | −1.657 | 0.0996 | . |
| fadn = C | −0.09261 | 0.327823 | −0.283 | 0.7779 | |
| fadn = D | −0.80073 | 0.426971 | −1.875 | 0.0626 | . |
| year = 2017 | −0.28932 | 0.156327 | −1.851 | 0.0661 | . |
| | | | log (ν) | | |
| Intercept | −1.5919 | 0.2288 | −6.959 | $9.08 \times 10^{-11}$ | *** |
| residue_colected = yes | −12.5212 | 435.7352 | −0.029 | 0.977 | |
| intercrop = yes | −1.0108 | 0.7677 | −1.317 | 0.19 | |

Cox Snell pseudo $R^2$ = 0.29; symbols ***, **, *, ` denote coefficients which are significant at 0.001, 0.01, 0.05, 0.1 levels, respectively.

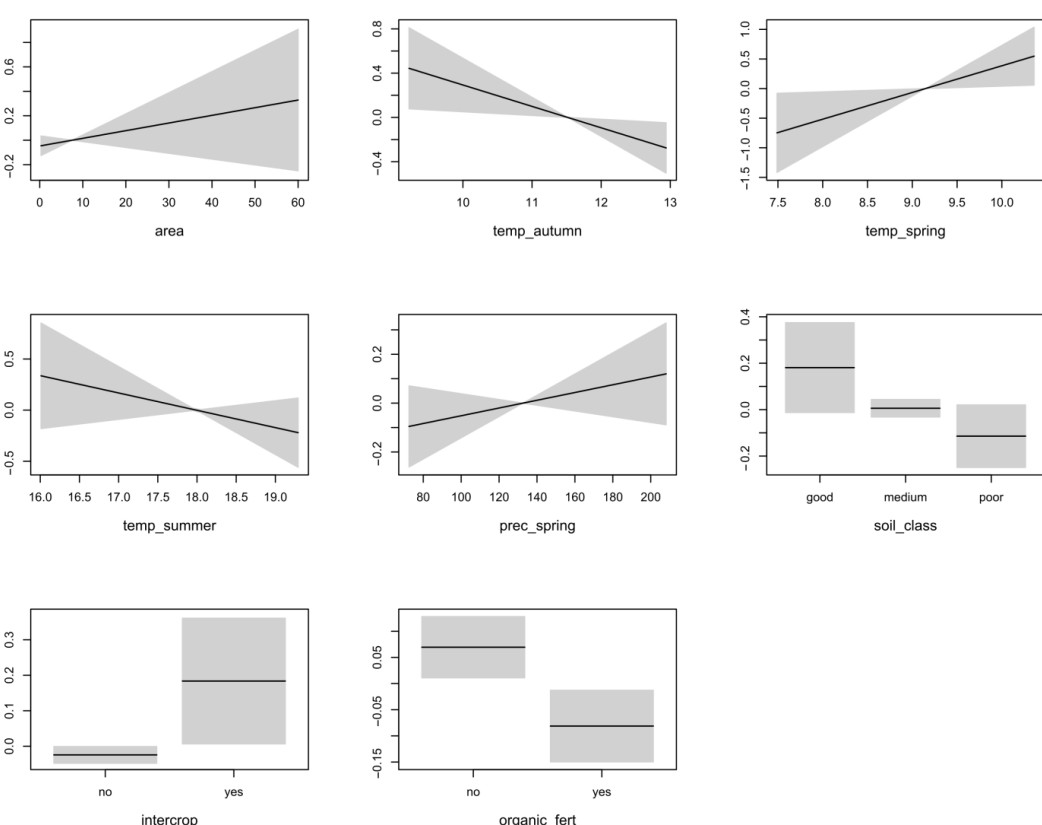

**Figure 3.** GAMLSS plots show the partial effects of explanatory variables on efficiency score (mu parameter) for cultivation of winter triticale. The *y*-axis represents the partial effect of each variable. The shaded areas indicate the 95% confidence intervals.

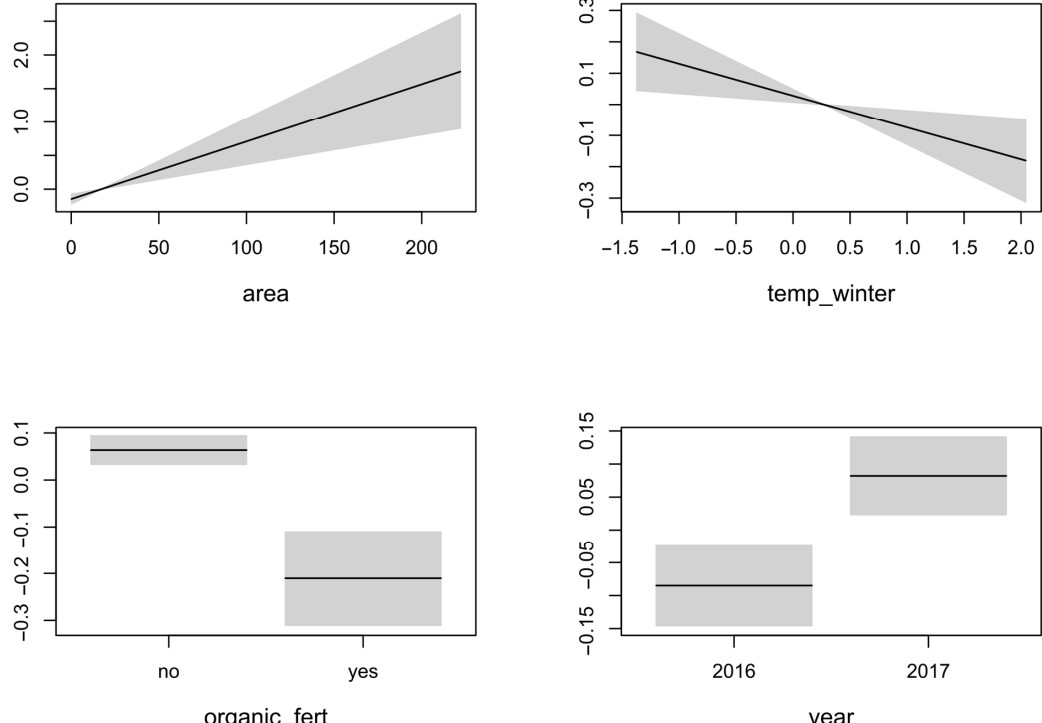

**Figure 4.** GAMLSS plots show the partial effects of selected explanatory variables on the mean value of efficiency score (mu parameter of BETAINF1 distribution) for cultivation of winter wheat. The *y*-axis represents the partial effect of each variable. The shaded areas indicate the 95% confidence intervals.

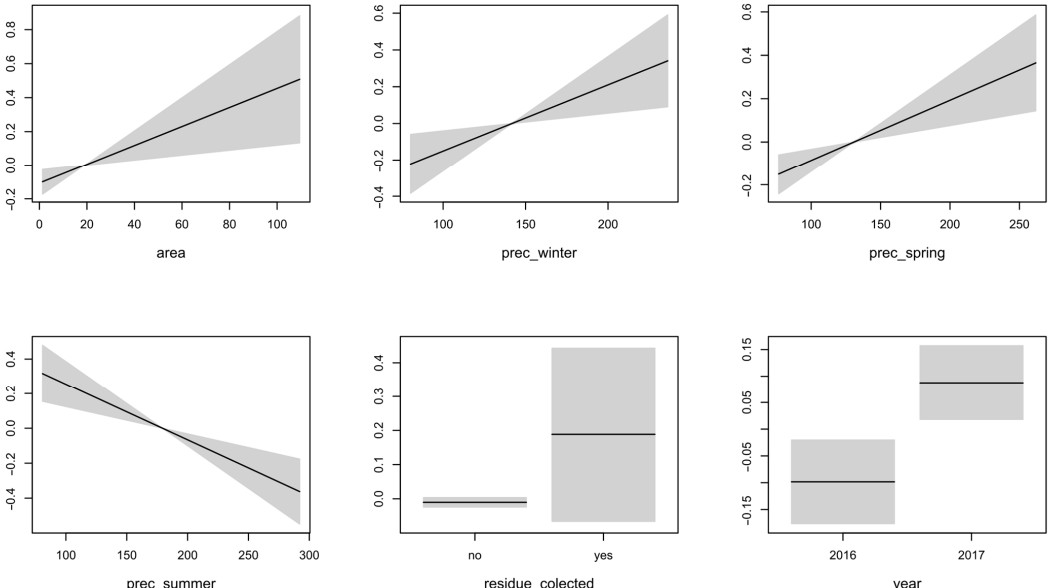

**Figure 5.** GAMLSS plots showing the partial effects of selected explanatory variables on the mean value of efficiency score (mu parameter) for cultivation of winter oilseed rape. The *y*-axis represents the partial effect of each variable. The shaded areas indicate the 95% confidence intervals.

The cultivation area positively affects the average PTE value for all studied crops (however, this parameter is not statistically significant for winter triticale cultivation). The largest effect was observed for winter wheat, followed by oilseed rape. This may be explained by the fact that triticale is grown both for own use in farm as a feed and for sale, whereas the other two crops are cash crops (triticale was most frequently grown on farms with a predominant pig production type (41%)) on an average area significantly

smaller than that of the other two crops. The area parameter has no significant effect on the variance (standard deviation) for either crop but increases the probability of being efficient for farms growing triticale and wheat (variables not present in the oilseed rape model).

Use of natural fertilizers negatively affected the average PTE value; the variable organic_fert appeared in the two final models (wheat and triticale). Oilseed rape is mainly grown on crop farms (70%), and organic nitrogen fertilization has the lowest contribution to its cultivation (comparing to other crops). Fertilization with organic fertilizers is fuel and time consuming, resulting in a lower efficiency score.

The weather patterns had a significant effect on the eco- efficiency of the investigated crops. A negative relationship between PTE score and autumn (for triticale) and winter temperature (for wheat) was confirmed by the models. Crop efficiency was positively influenced by spring rainfall (triticale and oilseed rape) and temperature (triticale). High precipitation in summer negatively effects eco-efficiency of winter rape, whilst high temperature in summer impacted negatively eco-efficiency of triticale.

The achieved results have not confirmed the dependence of PTE on the economic size of the farm and its type, as was obtained in similar works. The year variable affected the magnitude of efficiency for wheat and oilseed rape. This variable is a proxy for changes between years impossible to capture otherwise.

## 4. Discussion

The results of study indicate that the largest GHG emissions per hectare, as a measure of cultivation intensity, is caused by oilseed rape (4.3 $tCO_2e$), followed by wheat (3.2 $tCO_2e$) and triticale (1.9 $tCO_2e$). The carbon footprint of 1 kg of yield obtained in this study is not substantially different from the values presented in the meta-analysis by Clune et al. (2017), where the emissions associated with cultivation were estimated to be 0.53 ($\pm$0.22) kg $CO_2$ $kg^{-1}$ and 1.46 ($\pm$3.70) kg $CO_2$ $kg^{-1}$ for cereals and oilseed rape, respectively [49]. More than three-quarters of greenhouse gas emissions are associated with the production and use of nitrogen fertilizers, which is consistent with Hiller et al. (2009) [50], but also shows that emission reductions must be based on raising farmers' awareness of rational fertilizer use.

The economic size of the farms influenced the GHG emissions per hectare of cereals; in general, larger farms use more inputs (and emit more GHG) while achieving higher yields. A similar relationship was not found for oilseed rape. However, no differences were found in the carbon footprint of 1 kg of yield in relation to economic size and farm type.

The efficiency scores indicate a high potential for reducing resource use and GHG emissions. Efficient farms already use 14, 22, and 29% less fertilizers and achieve higher yields of 21, 23, and 14% for winter wheat, oilseed rape, and triticale, respectively.

It has also been shown that only a small percentage of farms operate at the right scale; it should be noted that the average areas of cultivated fields for the efficient farms were 8.2 ha, 16.4 ha and 17.6 ha for triticale, wheat and oilseed rape, respectively.

The achieved results have not confirmed the dependence of PTE on the economic size of the farm and its type, as was obtained in similar works. The study of Bieńkowski et al. (2019), who analyzed the eco-efficiency of winter triticale cultivation in two regions of Poland, showed that crop farms obtained higher environmental efficiency scores than pig and dairy farms [51]. However, the results presented at his work are not based on a statistical test, but on a comparison of the observed differences. The effect of the presence of animals on the efficiency score was investigated by Gutiérrez et al. (2017) using fractional regression, and a negative correlation was found [48], which is also not supported by the results we obtained.

A positive effect of cultivation's area on efficiency score (PTE) for all crops studied was confirmed. This is in line with the results of Zhang et al. (2021), which showed that an increase in farm size leads to a reduction in inputs and GHG emissions [52] and Kaditi et al. (2010), showing a positive effect of farm size on the efficiency score [48]. However, it should be noted that Ricciardi et al. (2021) in their meta-analysis found no clear difference between small or large farms in technical efficiency and GHG emission. Simulta-

neously, it was shown that the relationship between technical efficiency, greenhouse gas emissions, and farm size (measured in hectares) is spatially heterogeneous and country-specific. For example, in countries with a predominance of small family farms (for example India), those farms are more efficient, while in countries with a predominance of large farms (for example USA), they use resources more efficiently [53].

The results also confirmed the influence of other factors on eco-efficiency score. The positive effect of temperature and rainfall was found in spring season; an inverse relationship was shown for summer. Galushko and Gamtessa (2022) examined the effects of temperature and precipitation on the technical efficiency of Canadian crop production using a panel stochastic frontier model. They showed that the combined effect of higher temperatures and lower precipitation reduces TE. It was also shown that weather shocks, measured as deviations from historical weather averages, negatively affect technical efficiency [54]. The importance of adapting agrotechnical practices and plant varieties to climate change should be also pointed out. Practices (adjusting sowing dates and other climate adaptation treatments) and plant varieties that are more tolerant to water shortages and resistant to high temperature and disease should be used [2,55–57]. An integrated crop production system consisting of rational fertilization, the use of plant protection products and appropriate crop rotation can improve farm efficiency [58,59]. The use of natural fertilizers was revealed to have a negative impact on efficiency, mostly due to the higher consumption of fuel and machinery during this treatment; there is, therefore, a need to improve agrotechnics in this aspect.

However, attempts to explain the causes of inefficiency suffer from a number of limitations. Structural, organizational, and meteorological parameters were accounted for investigating their impact on the eco-efficiency score. The wide range of determinants of efficiency used has not provided fully satisfactory answers to the reasons for inefficiencies. Future research could be devoted to the inclusion of other environmental and socio-economic variables, such as farmer's experience, education level, participation in CAP instruments, and other factors.

### 5. Conclusions

The paper estimates the carbon footprint and cultivation's efficiency and of three crops with the largest area in Poland, namely: winter wheat, winter triticale, and winter oilseed rape.

Our findings indicate GHG emissions depend mainly on the amount of nitrogen fertilizer applied, as evidenced by high correlation scores (82–86%). Therefore, the reduction of greenhouse gas emissions must be based on the rational use of fertilizers. This includes improving nutrient control through management plans, using innovative approaches to minimize nutrient release, ensuring optimal pH and precision farming, and it is integrated into CAP policy [2].

No evidence was found that an economic size and farm type affect the carbon footprint of yield, while it was shown that input levels and GHG emissions per hectare vary across those factors.

The results suggest that the majority of the farms studied are not growing crops efficiently. There is a high potential to reduce environmental impacts without reducing yields. It has been shown that fully efficient farms use fewer inputs, especially fertilizer (14–29%) and fuel (17–25%), which translates into an average reduction in GHG emissions of 348–540 kg $CO_2$e ha$^{-1}$.

Another important implication is that resource-use efficiency is not affected by main structural factors; the influence of the economic size and farm type on efficiency score was not confirmed. In contrast, an increase in the size of the area under cultivation had a positive effect for all crops.

Weather patterns significantly affect the efficiency score; higher temperatures and rainfall during the intensive growing season (spring) are positively related to efficiency, while the opposite effect is observed in summer. It should be noted here that it is important

to promote crop varieties that are more tolerant to drought, higher temperatures, and more resistant to various diseases.

Finally, the results confirm that increasing the efficiency of crop cultivation can contribute to reducing greenhouse gas emissions and meeting the Green Deal targets.

**Author Contributions:** Conceptualization, T.Ż. and J.K.; methodology, T.Ż.; software, T.Ż.; writing—original draft preparation, T.Ż.; writing-review and editing, T.Ż. and J.K.; visualization, T.Ż.; funding acquisition, J.K. All authors have read and agreed to the published version of the manuscript.

**Funding:** Research was funded by Polish Ministry of Agriculture and Rural Development, as part of task DC 2.4. "Analysis of selected CAP instruments and their potential to reduce greenhouse gas and air pollutant emissions".

**Institutional Review Board Statement:** Not applicable.

**Informed Consent Statement:** Not applicable.

**Data Availability Statement:** Anonymized data available on request.

**Acknowledgments:** The authors would like to thank Małgorzata Wydra for the linguistic revision of the article.

**Conflicts of Interest:** The authors declare no conflict of interest.

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
