# Peer review of "Crop Cultivation Efficiency and GHG Emission: SBM-DEA Model with Undesirable Output Approach"

_sustainability, doi:10.3390/su151310557_

Round 1

Reviewer 1 Report

Overall, this is a well-written paper but needs some clarifications in the Abstract, introduction, and Materials Methods. The abstract should have some quantified data which describes major findings. The introduction should have a clear message about the needs of the research which is missing in the final part of the introduction but the importance of methodology is overall well written in the introduction. Materials and methods are too long which should be shortened for avoiding unnecessary detail.

Results and Discussion are well-composed and well-explained but have minor issues and confusion in English. These should be resolved to make results clear and understandable.

Thanks

Results and Discussion are well-composed and well-explained but have minor issues and confusion in English. These should be resolved to make results clear and understandable.

Reviewer 2 Report

The title clearly states the focus of the study, indicating that it examines crop cultivation efficiency and greenhouse gas (GHG) emissions using a specific approach. The abstract provides a concise summary of the study, including the data source, methods used, key findings, and the significance of the research. The study aims to estimate the carbon footprint of three major crops in Poland and evaluate the eco-efficiency of cultivation. It seeks to determine the factors influencing efficiency scores and explore the relationship between farm characteristics and performance. The study employs a slack-based Data Envelopment Analysis model (SBM-DEA) with an undesirable output (GHG emissions). This method allows for the evaluation of eco-efficiency by considering both desirable outputs (e.g., yield) and undesirable outputs (e.g., GHG emissions). Additionally, the Generalized Additive Model for Location, Scale, and Shape methods (GAMLSS) is used to explain efficiency scores. The study utilizes survey data from 250 farms, which provides a substantial sample size for analysis. It is important to assess the representativeness of the farms and consider any potential biases in the data collection process.The study indicates that most farms can improve their performance in terms of eco-efficiency. The fully efficient farms demonstrate lower usage of fertilizers, fuel, and pesticides while achieving higher yields and emitting fewer GHGs per hectare. The findings suggest a lower carbon footprint per kilogram of yield (ranging from 20% to 32%) for the examined crops. Additionally, the study highlights the impact of weather patterns on eco-efficiency. The study discusses the implications of the results, such as the potential for reducing GHG emissions and improving eco-efficiency in crop cultivation. It notes the importance of factors like cultivation area size and weather patterns. The absence of conclusive evidence regarding the relationship between economic size or farm type and performance is also mentioned. The provided keywords (GHG emission, eco-efficiency, crop cultivation) accurately reflect the main themes and concepts covered in the study. Please review my line-by-line comments and suggestions, especially grammatical suggestions, prior to publication.

Abstract

..to estimate the carbon footprint of three…

..cultivation, a slack-based..

..step, the Generalized Additive Model for Location, Scale, and Shape..

..less fertilizer, fuel, and pesticides and, at the same time, achieve….

..20–32% ..

…depending on the crop. It has also been shown..

..between the economic size…

..impacted the eco-efficiency of the analyzed crops…

The provided method titles correspond to important concerns in the study.

  1. Title: Data and data curation
    • Concern: Data collection and curation process
    • Content: This section describes the data used in the study, including the source and classification of the surveyed farms. It also explains the data curation process, specifically the use of the DBSCAN algorithm for detecting outlier observations. The result of the data curation process is presented in Table 1.
  2. Title: 2.2. Carbon footprint evaluation
    • Concern: Carbon footprint calculation
    • Content: This section explains the methodology for calculating the carbon footprint. It describes the scope of the analysis, including upstream emissions and direct on-farm emissions. It also provides information on emission factors and assumptions used in the calculations. A summary of input emission factors is presented in Table 3.
  3. Title: 2.3. Efficiency measurement using SBM-DEA model with undesirable output
    • Concern: Efficiency measurement using SBM-DEA model
    • Content: This section introduces the SBM-DEA model with undesirable output as the method used for efficiency measurement. It explains the parameters of DEA models, the orientation, assumption of returns to scale, and the advantages of the SBM-DEA approach. The specific inputs and outputs used in the model are described, and the deaR package is mentioned as the tool used for the calculations.
  4. Title: 2.4. Explaining efficiency
    • Concern: Explaining the causes of efficiency/inefficiency
    • Content: This section introduces the GAMLSS (Generalized Additive Model for Location, Scale, and Shape) method as the approach used to determine the causes of efficiency. It explains the choice of distribution for the dependent variable, the one-inflated beta distribution (BEINF1), and the estimation of GAMLSS models. It also describes the independent variables used in the models and the statistical tests used for comparison and significance analysis.

Please review my line-by-line comments and suggestions, especially grammatical suggestions, prior to publication in the method section:

Title: Data and data curation

..The data used in the study comes from surveys conducted in the 2015/2016 and 2016/2017 seasons on 250 farms across Poland….

.. according to the Farm Accountancy Data Network (FADN) classification: arable crops, dairy cows, and pigs…

…mineral and organic fertilizer application, pesticide use, seeds, and yield….

..The contents of nitrogen, phosphorus, and potassium..

..used on the farms were disaggregated by type of fertilizer..

…fertilizer equivalents according to Polish legislation [26].

To detect outlier observations in multivariate data, the density-based spatial clustering with noise (DBSCAN) algorithm proposed by Ester [14, 27], implemented in the dbscan package [28], was used.

..winter wheat, and winter oilseed rape, respectively.

This enabled the identification of outlier observations in low-density areas in multivariate data and did not require..

4.2% (13 farms), and 4.4% (….

..after data curation is presented in Table 1.

The analyzed farms were classified according to the Polish FADN..

..small (8–25 thousand EUR), medium (25–100 thousand EUR), and large (100–500 thousand EUR).

…climate data operator (CDO) software [30].

Title: 2.2. Carbon footprint evaluation

A carbon footprint..

..(CF_kg), were adopted for..

..missions (input production) as well as..

…nitrogen forms (nitrate, ammonium, ammonium-nitrate, and amide) was…

..yield (straw), and the nitrogen..

..or N2O, the global warming potential over a 100-year time horizon is equal to..

..for tractors, agricultural machines, and combine harvesters, according to..

..respectively: 29 kW for

(30–64 kW),..

(65–94 kW), and farm area > 80 ha

..carbon footprint of triticale seeds..

that of wheat. In accordance with..

..and the Biograce project,

..livestock production and has been assumed to be..

..matter changes, the balance..

Title 2.3. Efficiency measurement using SBM-DEA model with undesirable output

..over the years since Charnes et al. (1978)

..and the assumption of returns to scale. Input-oriented basic models aim to minimize inputs

output-oriented models aim to maximize output levels while producing

The CCR model measures the technical efficiency (TE) of a DMU under the constant return to scale (CRS) assumption, while efficiency under the variable..

..calculated by scaling efficiency (SE) as..

while the assumption of variable returns to scale (VRS) enables some of the DMUs to have a constant, decreasing, or increasing..

when a proportional increase in all the inputs results in a more than proportional increase in output, while decreasing return to scale (DRS) means a proportional..

In this study, an SBM-DEA input-oriented..

hypothesis of proportional variable changes.

pure technical, and scale efficiency

phosphorus, and potassium from mineral and natural fertilizers) [kg], diesel fuel [l], pesticides [kg a.i.], seeds [kg], and machinery [minutes].

in further model estimation due to the fact that increasing inputs in crop cultivation do not necessarily..

Title: 2.4. Explaining efficiency

..the generalized additive model for location, scale, and shape (GAMLSS),..

The GAMLSS model allows fitting..

(i.e., continuous and discrete) distribution that is continuous within the interval..

The study used a one-inflated beta distribution (BEINF1), which allows..

variable selection procedure based on the Generalized..

the procedure starts with an empty model (containing only constants)..

The GAMLSS package [48]..

To assess the impact of the habitat, quantitative variables describing the weather for the seasons of the year (temperature, precipitation) were used. A dummy variable year is used as a proxy for …

..as a proxy for differences resulting from tradition and the agricultural structure as well as input intensity. The effect of the use of organic fertilizers on..

In the following expression, ‘significant differences’ refers to test results for which the p-value…

These methodologies are widely used in the agricultural and environmental research fields to evaluate performance and investigate the impact of determinants. The article provides a clear description of the data collection and curation process, carbon footprint evaluation, and the application of SBM-DEA and GAMLSS, including the underlying assumptions.

However, it is important to note that the article lacks a dedicated discussion section, which limits the thorough examination and interpretation of the results. To address this limitation, integrating relevant findings from the results section and key points from the conclusion section is crucial for constructing a comprehensive and informative discussion. By doing so, a cohesive narrative can be developed, allowing for a deeper understanding of the implications of the study's findings.

Furthermore, the study primarily focuses on physical inputs and outputs, neglecting the consideration of other influential factors that could impact eco-efficiency, such as socioeconomic variables and farmer characteristics. This narrow focus may result in an incomplete understanding of the determinants of eco-efficiency in agricultural practices. It is recommended that future research endeavors incorporate a more comprehensive set of variables to enhance the understanding of the underlying factors influencing eco-efficiency.

In addition, it is vital to compare the study's findings with existing literature to evaluate their alignment or divergence from previous studies conducted in different regions or utilizing different methodologies. Such comparisons enable researchers to identify the study's contribution and novelty within the field. By examining the consistency or discrepancies with prior research, the results can be contextualized, and their broader implications can be better understood.

In conclusion, while the study's methodologies, namely the SBM-DEA model with undesirable output and GAMLSS, provide valuable insights into eco-efficiency and efficiency scores, it is crucial to incorporate a separate discussion section that considers limitations, such as the exclusion of additional influential factors, and facilitates comparisons with existing literature. By doing so, a comprehensive interpretation of the results can be achieved, enhancing the overall scientific rigor and impact of the study.

minor revision is requested

Reviewer 3 Report

Dear authors,

Please find attached here my comments and suggestions.

Best regards,

Reviewer 4 Report

The manuscript “Crop cultivation efficiency and GHG emission: SBM-DEA with undesirable output approach” used survey data from 250 farms to estimate carbon footprint of three crops with the largest sown area in Poland while using slack-based Data Envelopment Analysis model (SBM-DEA) for assessing the eco-efficiency of cultivation. Efficient use of farm inputs directly improves the farm efficiency and ultimately reduces the GHG’s. Farm size and weather conditions are important factors influencing the farm output and GHG’s. The MS is interesting for the readers. Some suggestions are given below to improve the readability of MS.

·     Title

Title “Crop cultivation efficiency and GHG emission: SBM-DEA with undesirable output approach” is confusing because have an abbreviation. Commonly it is avoided to use the abbreviations in title except common words. I think you may change the title as “Crop cultivation efficiency and GHG emission: SBM-DEA model with undesirable output approach”.

·     Abstract

Page 1, line # 8-9 “indicates that to estimate carbon footprint of three crops with the largest sown area in Poland” please write down the name of those crops and also try to indicate the efficiency based on crop type if possible.

·     Introduction

Introduction is looking fine.

·     M & M

The M&M is well written and explained the model very well, but the emissions are based on assumptions without any ground truthing, how will you justify the reliability of emission data or output or carbon foot printing while using literature data. Another Question is about the efficiency of SBM-DEA model in determining the eco-efficiency of various agricultural systems rather than cultivation. You mentioned in M&M that 250 farms were surveyed across Poland and these surveyed farms belonged to three types of production i.e. arable crops, dairy cows and pigs according to Farm Accountancy Data Network (FADN) classification while in abstract you mentioned only three types of crops, please clarify this as well and also try to synchronize title, abstract, introduction and M&M.

·     Results

It not only results but this section is indicating both results and discussions, so please change the heading. In modern word it is preferred to keep both results and discussion section separate for easy understanding of the readers. If possible, separate the sections and try to focus on results without arguments in result section but arguments should be in the discussion section.  

·       Conclusion

Conclusion is indicated as Section # 5 instead of Section # 4 because results are indicated as #3, so please correct it.

Conclusion is too long; I would always prefer to keep conclusion short showcasing best output of the study. This will keep the interest of reader and easy to get take home message.

All the best!   

-

Round 2

Reviewer 3 Report

Dear!

The authors addressed all my questions. The paper is now clear and well organized. from my point of view the article can be accepted in the present form

Warm regards!

Author Response

Dear Reviewer,

Thank you for your comments and suggestions .

Kind regards,

Tomasz Żyłowski